# Locally Secreted Semaphorin 4D Is Engaged in Both Pathogenic Bone Resorption and Retarded Bone Regeneration in a Ligature-Induced Mouse Model of Periodontitis

**DOI:** 10.3390/ijms23105630

**Published:** 2022-05-18

**Authors:** Takenobu Ishii, Montserrat Ruiz-Torruella, Kenta Yamamoto, Tsuguno Yamaguchi, Alireza Heidari, Roodelyne Pierrelus, Elizabeth Leon, Satoru Shindo, Mohamad Rawas-Qalaji, Maria Rita Pastore, Atsushi Ikeda, Shin Nakamura, Hani Mawardi, Umadevi Kandalam, Patrick Hardigan, Lukasz Witek, Paulo G. Coelho, Toshihisa Kawai

**Affiliations:** 1Department of Orthodontics, Tokyo Dental College, Tokyo 101-0061, Japan; ishiit@tdc.ac.jp; 2Institute Sant Joan, Carrer de Sant Joan, 26, Rubí, 08191 Barcelona, Spain; montseruizt@gmail.com; 3Department of Dental Medicine, Graduate School of Medical Science, Kyoto Prefectural University of Medicine, Kyoto 602-8566, Japan; fiori30@koto.kpu-m.ac.jp; 4Research and Development, LION Corporation, Odawara 256-0811, Japan; tsuguno@lion.co.jp; 5Department of Oral Science and Translational Research, College of Dental Medicine, Nova Southeastern University, 3200 South University Drive, Davie, Fort Lauderdale, FL 33328, USA; aheidari@nova.edu (A.H.); rp1258@nova.edu (R.P.); eleon@nova.edu (E.L.); sshindo@nova.edu (S.S.); mrawasqa@nova.edu (M.R.-Q.); mpastore@nova.edu (M.R.P.); snakamur@nova.edu (S.N.); 6Department of Periodontics and Endodontics, Okayama University Hospital, 2-5-1 Shikata-cho, Kita-ku, Okayama 700-8525, Japan; aikeda.0429@okayama-u.ac.jp; 7Department of Oral Diagnostic Sciences, Faculty of Dentistry, King Abdul-Aziz University, Jeddah 21589, Saudi Arabia; hmawardi@kau.edu.sa; 8Woody L. Hunt School of Dental Medicine, Texas Tech University Health Sciences Center El Paso, El Paso, TX 79905, USA; umadevi.kandalam@ttuhsc.edu; 9Patel College of Allopathic Medicine, Nova Southeastern University, 3200 South University Drive, Davie, Fort Lauderdale, FL 33328, USA; patrick@nova.edu; 10Division of Biomaterials, NYU College of Dentistry, New York, NY 10010, USA; lw901@nyu.edu (L.W.); pc92@nyu.edu (P.G.C.); 11Cell Therapy Institute, Center for Collaborative Research, Nova Southeastern University, 3200 South University Drive, Fort Lauderdale, FL 33328, USA

**Keywords:** periodontitis, Semaphorin 4D, bone regeneration, osteoblasts, osteoclasts

## Abstract

It is well known that Semaphorin 4D (Sema4D) inhibits IGF-1-mediated osteogenesis by binding with PlexinB1 expressed on osteoblasts. However, its elevated level in the gingival crevice fluid of periodontitis patients and the broader scope of its activities in the context of potential upregulation of osteoclast-mediated periodontal bone-resorption suggest the need for further investigation of this multifaceted molecule. In short, the pathophysiological role of Sema4D in periodontitis requires further study. Accordingly, attachment of the ligature to the maxillary molar of mice for 7 days induced alveolar bone-resorption accompanied by locally elevated, soluble Sema4D (sSema4D), TNF-α and RANKL. Removal of the ligature induced spontaneous bone regeneration during the following 14 days, which was significantly promoted by anti-Sema4D-mAb administration. Anti-Sema4D-mAb was also suppressed in vitro osteoclastogenesis and pit formation by RANKL-stimulated BMMCs. While anti-Sema4D-mAb downmodulated the bone-resorption induced in mouse periodontitis, it neither affected local production of TNF-α and RANKL nor systemic skeletal bone remodeling. RANKL-induced osteoclastogenesis and resorptive activity were also suppressed by blocking of CD72, but not Plexin B2, suggesting that sSema4D released by osteoclasts promotes osteoclastogenesis via ligation to CD72 receptor. Overall, our data indicated that ssSema4D released by osteoclasts may play a dual function by decreasing bone formation, while upregulating bone-resorption.

## 1. Introduction

Periodontal diseases (PD) is one of the most prevalent diseases in humans, affecting 42.2% of the United States population with 7.8% of people experiencing severe periodontitis [1]. Moreover, an increasing number of reports support PD as a risk factor for many systemic diseases [2,3]. PD is characterized by chronic inflammation of tooth-supporting tissue, which leads to connective tissue destruction and alveolar bone resorption [4,5]. The etiology of multifaceted PD remains elusive, especially in light of unclarified etiopathological differences between PD and gingivitis, noting that gingivitis involves inflammation in the periodontium without causing bone resorption [6]. We know that host immune response to periodontopathic bacteria likely involves the recruitment of inflammatory cells that trigger an inflammatory cascade accompanied by abundant tissue damage, including the bone resorption seen in PD [7,8,9,10,11]. We also know that the progression of PD causes irreversible bone loss mediated by pathogenically promoted osteoclastogenesis via local production of receptor activator of NF-kB ligand (RANKL) by bacterial reactive immune cells [9,12,13].

Homeostatic bone remodeling consists of bone catabolic activity (bone resorption) mediated by osteoclasts, followed by bone anabolic activity by osteoblasts. Bone regenerative factors that stimulate the anabolic activity of osteoblasts are released during bone resorption [14,15]. This stimulation of osteoblast activity in response to osteoclast-mediated resorption is termed “coupling” [16], which is believed to mediate the homeostatic remodeling of the healthy bone. However, consensus supports that the coupling mechanism is dysregulated in the context of diseased bone, in particular, periodontitis [17]. This explains, in part, why bone regeneration rarely occurs in the periodontal tissue affected by periodontitis. Currently, the molecular mechanism underlying irreversible bone loss observed in periodontitis remains to be elucidated.

Semaphorin 4D (Sema4D), also known as CD100, is a cell-surface protein originally identified as expressed in immune cells [18]. Belonging to the Semaphorin family of secreted, membrane-bound proteins [19,20], Sema4D binds to three different receptors, including the high-affinity receptor Plexin-B1 and the low-affinity receptors CD72 and Plexin-B2 [21,22]. Sema4D is a multifunctional molecule [21,23] in that it has distinct roles in immune response [24,25], axonal guidance [26,27], and angiogenesis [28]. More recently, it has also been reported that Sema4D plays an important role in bone tissue. In brief, Sema4D-knockout mice showed an increase in bone thickness and density in a mouse model of osteoporosis [29], and Sema4D expressed on osteoclasts suppresses osteoblast differentiation via receptor PlexinB1 [30,31]. Importantly, an elevated level of Sema4D protein was detected in the gingival crevice fluid (GFC) of patients with PD compared to that from periodontally healthy subjects [32]. Furthermore, it is reported that the serum Sema4D level in rheumatoid arthritis and postmenopausal osteoporosis patients was higher than that in healthy subjects [33,34]. The effects of alternative splicing on transcripts encoding membrane proteins has been investigated in the context of the production of novel soluble protein isoforms. While two alternative splicing of human Sema4D were reported (UniProtKB: Q92854-1 and Q92854-2), according to the deduced protein sequences, those are still produced in membrane bound form. Thus, it is theorized that the identification of a soluble form of Sema4D detected in the GCF of patients with PD [32], as well as serum Sema4D found in rheumatoid arthritis and postmenopausal osteoporosis [34], results from cleaving of membrane-bound Sema4D expressed on the cell surface [35]. Soluble Sema4D seems to be produced in the context of inflammatory lesions but not in normal tissue without inflammation [35]. Additionally, our group recently discovered that Sema4D expressed on osteoclasts is cleaved by tumor necrosis factor-alpha converting enzyme (TACE; also called ADAM17) (paper under review), which plays a relevant proinflammatory role in periodontitis [36]. We also demonstrated that soluble Sema4D released from activated platelets can promote osteoclastogenesis in vitro [37]. Nonetheless, the possible role of Sema4D produced in periodontitis lesions, which is perhaps produced in a soluble form, remains elusive. The present study examined the functional role of Sema4D in osteoclastogenesis induced in a mouse model of ligature-induced periodontitis.

## 2. Results

### 2.1. Increased Production of Sema4D in Mouse Periodontal Tissue of Experimentally Induced PD

To monitor the expression of Sema4D in periodontal tissue with or without PD, a silk ligature was attached to the second maxillary molar of C57BL/6J mice. Seven days after ligature attachment, significantly elevated bone resorption around the tooth was observed (Figure 1A). Soluble RANKL (sRANKL), soluble Sema4D (sSema4D) and IGF-1 were all elevated in the gingival tissue induced by PD compared to that of the control healthy group (Figure 1B). However, the attachment of the ligature to the second maxillary molar of mice did not affect the production of sSema4D in the femur (Figure 1B). Moreover, Sema4D-expressing cells were prominent on the alveolar bone surface of the PD group compared to that of the control group in periodontal tissue collected at Day 7 (Figure 1C), suggesting that those Sema4D-positive cells may be the source of sSema4D detected in the periodontal tissue induced by PD.

### 2.2. Characteristics of Anti-Sema4D mAb in Reaction to Sema4D Produced by Osteoclasts

In the in vitro M-CSF/RANKL-induced osteoclastogenesis assay using BMMCs as osteoclast precursors (Figure 2A: TRAP and pit formation assays), increased expression of mRNA for Sema4D, as well as OC-STAMP, NFATc1 DC-STAMP and ATP6v0d2, by RANKL addition was detected by PCR (Figure 2B). The Sema4D produced by osteoclasts downregulated IGF-1-induced osteoblast differentiation [30]. For the loss-of-function analysis of Sema4D, we generated an anti-Sema4D neutralizing mAb [38]. According to W-blot analysis, in response to stimulation with M-CSF/RANKL for 24, 48 and 72 h, increased expression of a monomeric membrane-bound form (150 kD) and a dimeric membrane-bound form (300 kD) was observed with the highest level detected after a 48-hour incubation in the cell homogenates prepared under nonreducing conditions (Figure 2C), corresponding to a previous report [39]. ELISA showed a significantly elevated production of soluble Sema4D (sSema4D) in the culture supernatant of BMMCs stimulated with RANKL/M-CSF for 24, 48 and 72 h (Figure 2D. MC3T3-E1 cells incubated with Vit C and β-GP (OB-supplement) for 7 days showed elevated ALP production compared to control MC3T3-E1 cells without OB-supplement (Figure 2E). However, MC3T3-E1 cells stimulated with OB-supplement in the presence of supernatant harvested from the RANKL/M-CSF-primed osteoclast precursor cells (48 h) showed a significantly lower level of ALP expression when compared to cells cultured only with OB-genesis supplement. Such suppressive effect by the supernatant of RANKL/M-CSF-primed osteoclast precursor cells was abrogated by the addition of anti-Sema4D mAb to OCgenesis, but not control mAb (Figure 2E). An expression pattern similar to that of ALP staining was also observed in MC3T3-E1 cells stained for Alizarin Red (Figure 2E). These results indicated that RANKL/M-CSF-primed osteoclast precursor cells could produce functionally active sSema4D able to suppress the OB-genesis, but that anti-Sema4D mAb could neutralize such effect.

It is well known that periodontal bone loss induced in rodents is spontaneously regenerated after the removal of inflammatory stimuli, such as ligature [40,41]. However, we have employed the mouse model of ligature-induced periodontitis to determine the possible role of Sema4D in alveolar bone regeneration in PD. After attachment of a ligature for 7 days, it was removed from mice which then received either control mAb or anti-Sema4D mAb, followed by additional injections of Calcein Blue (Day 7) and Alizarin Red (Day 14). Control mice received only Calcein Blue (Day 7) and Alizarin Red (Day 14) without attachment of ligature or injection of mAb. As expected, the removal of ligature at Day 7, followed by systemic administration of control mAb, induced spontaneous bone regeneration accompanied by the increased *COL1a1* gene expression in the mice induced of PD (Figure 3B). In contrast, PD-induced mice that received systemic administration of anti-Sema4D mAb showed a significantly increased amount of alveolar bone regeneration as well as *COL1a1* gene expression compared to PD-induced mice that received control mAb (Figure 3B). suggesting that neutralization of Sema4D can promote alveolar bone regeneration in periodontal tissue affected by PD.

To determine the effect of xSema4D-mAb on systemic bone remodeling, fluorescent bone labeling, as well as micro-CT analysis, was performed on healthy mice (Figure 3C,D). Based on the fluorescent labeling, bone deposition on the cortical bone was not affected by xSema4D mAb (Figure 3D). No significant difference was noted between xSema4D-mAb- and control mAb-treated groups on the bone morphometry data obtained by micro-CT, including ratio of total bone volume (BV/TV), trabecular number (TbN), trabecular thickness (TbTh), and trabecular spacing (TbSp) (Figure 3D). Taken together, these results demonstrated that Sema4D may be engaged in downregulation of local osteoblastogenesis in PD, but not in healthy periodontal alveolar bone or peripheral bones.

### 2.3. Locally Elevated Expression of Sema4D in PD Lesion May Be Engaged in the Upregulation of Osteoclast-Mediated Periodontal Bone Loss

Systemic administration of anti-Sema4D mAb suppressed alveolar bone loss induced in mice by the attachment of the ligature (Figure 4A–C). According to the histology section stained with H&E, the inflammatory infiltrates increased in the ligature-induced periodontitis lesion irrespective of the administration with anti-Sema4D mAb (Figure 4D). In the homogenate of periodontal tissue isolated from mice induced of PD by attachment of ligature, the levels of TNF-α, sRANKL and IGF-1 were all significantly higher compared to control mice without ligature. No significant difference was noted in the levels of TNF-α, sRANKL, IGF-1 and IL-10 between the groups that received control mAb and xSema4D-mAb (Figure 4E), indicating little or no effect from the administration of anti-Sema4D mAb on local inflammatory response induced by onset of PD. To examine the effect of sSema4D on the immune cells, mouse bone marrow cells were stimulated with or without LPS in the presence or absence of Sema4D. Within the concentrations of sSema4D tested (0.1–1000 ng/mL), LPS-dependent-induced production of TNF-α, IL-6 and IL-1β by bone marrow cells were not affected by sSema4D (Appendix A). The level of OPG was suppressed by xSema4D-mAb (Figure 4E) which, in turn, increases the RANKL/OPG ratio. The IgG antibody response to mouse oral opportunistic pathogen, *Rodentibacter pneumotropicus* (formally termed as *Pasteurella pneumotropica*), was induced by the ligature attachment, which was also not affected by anti-Sema4D mAb (Figure 4F). Nonetheless, the PD-dependent increase in the number of TRAP-positive cells in alveolar bone was downmodulated by the systemic administration of anti-Sema4D mAb compared to control mAb (Figure 5A). The elevated expressions of osteoclastogenesis-related genes, including *ACP5* (TRAP), *CTSK* (Cathepsin K) and *Atp6v0d2*, were all suppressed by the systemic administration with anti-Sema4D mAb, while elevated *MMP9* mRNA was not affected by anti-Sema4D mAb (Figure 5B). These results indicate that locally elevated production of sSema4D in periodontal tissue, but not in the femur, of the mice receiving ligature attachment (Figure 1) may lead to the promotion of RANKL-induced osteoclastogenesis and resulting alveolar bone loss.

### 2.4. Sema4D Expressed by Osteoclasts Appeared to Facilitate the Upregulation of Osteoclastogenesis

As shown by the in vitro RANKL-induced osteoclastogenesis assay, the emergence of TRAP-positive multinuclear cells was significantly reduced in the presence of anti-Sema4D mAb compared to that of control mAb (Figure 6A). The activity of osteoclasts to resorb bone mineral was also suppressed by the addition of anti-Sema4D mAb, but not by control mAb in the pit formation assay performed on dentine slices (Figure 6B).

### 2.5. Sema4D Upregulates Osteoclastogenesis via Ligation with CD72 Expressed on Osteoclasts

Based on in vivo and in vitro experiments (Figure 4, Figure 5 and Figure 6A,B), we assumed that Sema4D produced in an autocrine fashion is engaged in the promotion of osteoclast differentiation and function. If this premise is true, osteoclast precursors should express the Sema4D receptor. Among three known receptors for Sema4D, i.e., Plexin B1, Plexin B2, and CD72 [35], Sema4D expressed on osteoclasts inhibits IGF-1-mediated osteogenesis by binding with PlexinB1 expressed on osteoblasts [30]. However, the receptor(s) involved in Sema4D-mediated osteoclastogenesis on osteoclasts precursor cells remain(s) largely unknown. Prominently elevated CD72 mRNAs were detected in BMMCs in response to RANKL/M-CSF stimulation. Although PlexinB2 mRNA was detected in unstimulated BMMCs at a level similar to that of CD72 mRNA, RANKL/M-CSF stimulation did not upregulate Plexin B2 mRNA expression (Figure 6C). In contrast, no detectable PlexinB1 mRNA was observed in BMMCs, regardless of stimulation with RANKL/M-CSF. The number of TRAP-positive multinuclear cells was significantly decreased in cells cultured with RANKL/M-CSF in the presence of anti-CD72 mAb compared to cells cultured with RANKL/M-CSF but was not changed in cells cultured with RANKL/M-CSF in the presence of control IgG or anti-Plexin B2 mAb (Figure 6D). These results indicated that expression of CD72 induced by RANKL on osteoclast precursor cells is the receptor of Sema4D, leading to upregulation of osteoclastogenesis.

## 3. Discussion

In this study, we demonstrated that Sema4D production was increased in periodontal tissue of ligature-induced PD and that systemic administration with anti-Sema4D mAb not only promoted bone regeneration, but also inhibited bone resorption in the context of PD. Anti-Sema4D mAb-mediated bone remodeling regulation was detected only in PD lesions, but not in homeostatic bone remodeling in the femur, suggesting a novel role of Sema4D in promoting osteoclastogenesis in the context of inflammatory bone resorption. Although the suppressive effect of Sema4D expressed by osteoclasts on IGF-1-mediated osteoblastogenesis was reported in the bone remodeling processes [30], no study, to the best of our knowledge, has demonstrated the role of autocrine Sema4D in osteoclastogenesis. One of the main differences between alveolar bone with PD and intact skeletal bone of the femur is the generation of soluble Sema4D. Although Sema4d–/– mice show the osteo-petrotic phenotype, in the skeletal bone of wild-type mice, only membrane-bound Sema4D, but not soluble Semaphorine 4D, is detected [30], indicating that cell–cell-contact-mediated ligation between Sema4D and Plexin B1 plays a role in Sema4D-mediated downregulation of osteoblastogenesis in homeostatic bone remodeling. Thus, this is the first finding that soluble Sema4D expressed by osteoclasts is engaged in their differentiation and resorptive function. In contrast to the inhibition of osteoblastogenesis mediated by ligation of Sema4D to PlexinB1, CD72 was revealed as the receptor for Sema4D in the Sema4D-mediated promotion of osteoclastogenesis.

In GFC of patients with PD, Sema4D level is significantly elevated compared to that from periodontally healthy subjects [32]. It was also reported that *Enterococcus faecalis*, one of the major pathogens implicated in periapical periodontitis, promotes periapical bone resorption via increasing Sema4D expression [42]. On the other hand, soluble Sema4D-producing γδ-T cells are related to pathogenesis of MRONJ [38]. These results, including our finding, suggested that Sema4D is engaged in the pathogenesis of bone remodeling disorders in the oral and maxillofacial regions. The role of sSema4D in promotion of pathogenic bone resorption was also reported in the mouse model of collagen-induced rheumatoid arthritis [34] as well as cancer bone metastasis [43]. Although, the latter two reports indicated that sSema4D mediated elevation of TNF-α production by monocytes as well as that of sRANKL production by osteoblasts are, in part, responsible for the increased osteoclastogenesis, respectively, [34,43], we could not find such an effect of Sema4D on the productions of TNF-α or sRANKL in the mouse periodontal tissue induced of periodontitis (Figure 4D). Such a discrepancy may be attributed to the diverse expression pattern of Sema4D receptors, including, Plexin B1, Plexin B2 and CD72 that elicits distinct cell signal in a variety of cells in respective pathologic context [24]. Future studies are needed to address the nature of receptors for soluble Sema4D expressed in periodontitis. 

Although we addressed the possible role of Sema4D in bone remodeling processes, the multifunctional molecule Sema4D is implicated in cancer angiogenesis, as well as axonal guidance during neuronal development [44,45,46]. Since robust microvasculature is required for the regeneration of resilient bone [47], further studies are needed to investigate pathological and physiological role of Sema4D in the angiogenic response in PD.

In recent years, several drugs have been made available to counteract pathogenic osteolysis. Among them, anti-bone resorptive agents, such as bisphosphonate and anti-RANKL antibody (Denosumab), markedly inhibit the pathogenic bone resorption in osteoporosis and cancer bone metastasis [48,49,50]. However, as the coupling of bone formation to resorption is a tightly regulated, treatment of osteolytic conditions with these anti-bone resorptive agents can result in the dysregulation of bone formation in the long term. For this reason, increasing lines of evidence support that these agents induce atypical femoral fracture [51] and medication-related osteonecrosis of jaw (MRONJ) [52,53]. These side effects derived from the administration of bisphosphonate or anti-RANKL antibody call for the development of a novel therapeutic modality able to regulate bone formation and bone resorption simultaneously. In this study, we demonstrated that Sema4D has dual functions of decreasing bone formation, while upregulating bone resorption. Interestingly, although Sema4D expression was elevated in PD induced in mice, anti-Sema4D mAb had no effect on homeostatic bone remodeling in femur (Figure 3D,E). In sum, the present study suggests the possible engagement of Sema4D in pathogenic bone resorption and retarded bone regeneration in the inflammatory bone lytic lesion of PD, but not in the healthy bone remodeling process. Therefore, Sema4D could be a novel molecular target for the development of a bone regenerative approach for PD without affecting systemic bone remodeling.

Sema4D functions as a ligand, mainly binding three different receptors. While the expressions of PlexinB1 and Plexin B2 are found on non-lymphoid cells, CD72 expression is limited on lymphoid cells. It is true that Sema4D derived from osteoclasts binds to PlexinB1-expressing osteoblasts which are non-lymphoid mesenchymal lineage cells [30]. On the other hand, CD72 is expressed on B cells and dendritic cells, both lymphocytes originated from hematopoietic stem cells. Nonetheless, CD72 functions as a receptor for Sema4D to promote the proliferation and cytokine secretion by those lymphocytes [54,55]. The present study demonstrated that autocrine Sema4D binds to CD72 expressed on osteoclast precursor cells which also belong to monocyte lineage cells derived from hematopoietic stem cells. In the 1990s, CD5 was initially reported to be the ligand for CD72 [56], even though this result could not be reproduced by another group [57]. Later, Semaphorin-4D, also known as CD100, was shown to be an inhibitory ligand of CD72 [58]. Owing to the unique property of CD72 that not only has immunoreceptor tyrosine-based inhibition motif (ITIM), but also works in concert with other membrane receptors, such as B cell receptor, the molecular mechanism underlying inhibitory signaling elicited by CD72 remains to be elucidated [59]. Thus, it is intriguing to elucidate the detailed cell signaling mediated by Sema4D ligation to CD72 during the osteoclastogenesis. In summary, we report the possible role of Sema4D in periodontal tissue that promotes osteoclast differentiation via ligation with CD72. These results may lay the groundwork for development of a novel therapeutic strategy for PD.

## 4. Materials and Methods

### 4.1. Animals

C57BL/6J mice were purchased from Jackson Laboratory and were bred in the animal facility at Forsyth Institute. Experimental procedures using mice were approved by the Institutional Animal Care and Use Committee (IACUC) at Forsyth Institute. Some experiments using bone marrow cells isolated from C57BL/6J mice were also approved by IACUC at Nova Southeastern University. This study was performed in accordance with ARRIVE guidelines for preclinical animal studies.

### 4.2. Mouse Bone Marrow Cells Culture and Osteoclast Differentiation Assay

Bone marrow-derived mononuclear cells (BMMCs) were established as previously descried [60,61]. Briefly, bone marrow cells isolated from the femur and tibia of C57BL/6J mice were seeded in a 96-well plate (1 × 10^5^ cells/well) in α-modified Minimal Essential Medium (α-MEM, Thermo Fisher Scientific, Waltham, MA, USA) supplemented with 10% fetal bovine serum (FBS, Atlanta Biologicals, Lawrenceville, GA: termed as “basal medium”). After pre-incubation of bone marrow cells in basal medium containing macrophage colony-stimulating factor (M-CSF, R&D Systems, Minneapolis, MN, USA) for 3 days, the tissue culture plate was washed with basal medium, and the remaining adherent cells in the plate were used as BMMCs. BMMCs were further cultured in basal medium supplemented with M-CSF (50 ng/mL) and RANKL (100 ng/mL) (BioLegend, San Diego, CA, USA) in the presence or absence of anti-Sema4D mAb (mouse IgG1 30 µg/mL) or control isotype matched mAb (IgG1, 30 µg/mL) [38]. In some experiments, RANKL-stimulated BMMCs were also cultured with or without anti-CD72 mAb (IgG1, 50 µg/mL), anti-Plexin B2 mAb (IgG1, 50 µg/mL), or control mAb (IgG1, 50 µg/mL [37]). Cultured cells expressing tartrate-resistant acid phosphatase (TRAP) were stained using the Acid Phosphatase Leukocyte kit (Sigma-Aldrich, Saint Louis, MO, USA) at Day 7 in accordance with the manufacturer’s instruction. TRAP-positive cells containing 3 nuclei or more were counted microscopically as osteoclasts. The bone resorption activity of osteoclasts was evaluated by pit formation assay using dentin disks (Alpco Diagnostics, Windham, NH, USA), following the protocol published previously [8].

### 4.3. Osteoblastogenesis Assay

MC3T3-E1 cells (mouse osteoblast precursors) were seeded in a 96-well plate (5 × 10^4^ cells/well) and cultured in basal medium with or without 50 μg/mL L-ascorbic acid (Vitamin C (Vit C); Sigma Aldrich, St. Louis, MO, USA) and 5 mM β-glycerophosphate (Sigma Aldrich) (osteoblast (OB) genesis supplement) in the presence or absence of supernatant harvested from the cultured osteoclast precursor cells with or without anti-Sema4D mAb (50 µg/mL) or control IgG (50 µg/mL). After 7 days of culture, the cells were stained with Alkaline Phosphatase (ALP) Staining kit (Sigma Aldrich) following the manufacturer’s instruction. The intensity of each well stained with ALP was measured using Image J software (version 1.53f25 software, Bethesda, MD, USA).

### 4.4. Real-Time RT-PCR

After stimulation of BMMCs with M-CSF or RANKL/M-CSF, total RNA was extracted from the cells using Trizol (Thermo Fisher) and were reverse transcribed using the Verso cDNA synthesis kit (Thermo Scientific) in the presence of random primers and oligo (dT)**.** The resulting cDNA was mixed with Real-time PCR Master Mix (Roche Diagnostics), SYBR Green (Roche Diagnostics) or Taqman fast advanced Master Mix (Thermo Fisher), and the primer set was subjected to gene expression analysis using the LightCycler 480 system (Roche Diagnostics). Amplification of the beta-actin or GAPDH gene was used as an internal control. The following primers were used in this study [37,60,61,62].

Sema4D Forward: 5′-TGATCCCTAGGTCAGACGGG-3′

Reverse: 5′-CTGGCTTGTGAAACTGCACC-3′

OC-STAMP Forward: 5′-ATGAGGACCATCAGGGCAGCCACG-3′

Reverse: 5′-GGAGAAGCTGGGTCAGTAGTTCGT-3′

NFATc1 Forward: 5′-CCTCGAACCCTATCGAGTGT-3′

Reverse: 5′-GCCAGACAGCACCATCTTC-3′

DC-STAMP Forward: TCCTCCATGAACAAACAGTTCCAA, R:

Reverse: AGACGTGGTTTAGGAATGCAGCTC

PlexinB1 Forward: 5′-CCCTCGGTCTCCGGGTAAG-3′

Reverse: 5′-CATGACCTGAGCAGGAGTCAC-3′

PlexinB2 Forward: 5′TGGTTCCTGCTGTAGCCATC-3′

Reverse: 5′-GATGTCTCCGTGCTTCCTGA-3′

CD72 Forward: 5′-CTGCACATCTCTGTCCTCCA-3′

Reverse 5′-TCAGAGTCCTGCCTCCACTT-3′;

β-actin Forward:5′-CTAAGGCCAACCGTGAAAAG-3′

Reverse: 5′-ACCAGAGGACTACAGGGACA-3′

ACP5: Mm00475698_m1 (Thermo Fisher)

Cathepsin K: Mm00484039_m1 (Thermo Fisher)

Atp6v0d2: Mm01222963_m1 (Thermo Fisher)

MMP9: Mm00442991_m1 (Thermo Fisher)

RUNX2: Mm00501584_m1 (Thermo Fisher)

GAPDH: Mm99999915_g1 (Thermo Fisher)

### 4.5. Enzyme-Linked Immunosorbent Assay (ELISA)

BMMCs were cultured with M-CSF/RANKL for 24 to 72 h, and the conditioned medium was harvested. The amount of Sema4D produced in the culture medium was measured by ELISA (RayBiotech, Peachtree Corners, GA, USA) following the method reported previously [38]. The level of serum IgG antibody reacting to mouse oral commensal, *Rodentibacter pneumotropicus* (formally termed as *Pasteurella pneumotropica*), was also determined using ELISA following the method published previously [62,63].

### 4.6. Western Blotting

Western blotting was performed following the methods published previously [62]. Briefly, after stimulation of BMMCs with M-CSF/RANKL for 24 to 72 h, cell lysates were centrifuged at 15,000 rpm for 15 min to remove insoluble components. The proteins in the lysates were separated by sodium dodecyl sulfate polyacrylamide gel electrophoresis (SDS-PAGE). After transferring to a nitrocellulose (NC) membrane, the NC membrane was reacted overnight at 4 °C with anti-mouse Sema4D-mAb (mouse IgG1) or control anti-mouse alpha tubulin-mAb (mouse IgG1, Abcam, Waktham, MA, USA). After overnight incubation of transblotted NC membrane at 4 °C with HRP-labeled anti-mouse immunoglobulin antibody, positive signals were visualized using the ECL Prime detection reagent (GE Healthcare, Chicago, IL, USA).

### 4.7. Ligature-Induced PD Model and Anti-Sema4D Antibody Injection

PD was induced in C57BL/6Jmice (6- to 8-week-old male, *n* = 6/group) according to the previously published protocol [62,64]. Briefly, silk ligatures (5-0; Ethicon) were placed in a subgingival position around the cervix of maxilla second molars in each animal, while no ligature was used on the counter side as control. At the same time, anti-Sema4D mAb (3 mg/mouse) or control IgG (3 mg/mouse) was administered via intraperitoneal injection (i.p.), following the protocol established for the mouse model of periodontitis that was used for the examination of anti-OC-STAMP and anti-DC-STAMP mAb’s efficacy in suppressing periodontal bone resorption [60,62]. The procedure was performed using two Castroviejo Micro Needle Holders under the stereomicroscope. After 14 days (endpoint), maxilla and gingival tissue were collected from scarified mice and evaluated.

### 4.8. Assessment of Alveolar Bone Resorption, Histological Analysis of Periodontal Tissue Section and ELISA of Tissue Homogenate

The maxillae of each mouse collected at the endpoint were defleshed and soaked in toluidine blue solution. Stained maxillae were photographed using a dissection microscope (Nikon), and total alveolar bone loss was calculated by measuring the cemento-enamel junction (CEJ) to the alveolar bone crest decalcification (ABC) distances on the buccal side of each root [41]. Some maxillae were fixed in 10% formaldehyde and then 10% EDTA solution at 4 °C for 3 to 4 weeks. The decalcified maxillae samples were then embedded in paraffin and sectioned (7-μm thickness) for TRAP and hematoxylin and eosin (H&E) staining. For immunofluorescence staining, the decalcified maxillae samples were embedded in OCT compound (Fisher Scientific) and sectioned (6 μm thickness) using a cryostat. The expression of Sema4D was monitored using biotin-conjugated anti-Sema4D-mAb, followed by avidin conjugated to AlexaFluor 488. Gingival tissue homogenates were obtained from the gingival tissue of each mouse at the endpoint, and Sema4D, TNF-α, RANKL, OPG, IL-10 and IGF-1 production in the homogenate was analyzed using an ELISA kit purchased from the following sources: Sema4D: RayBiotech; TNF-α, RANKL, OPG, IL-10 and IGF-1: R&D Systems.

### 4.9. Fluorescent Bone Labeling

Mice (6- to 8-week-old C57BL/6J male, *n* = 6/group) were induced with periodontitis by attachment of a ligature for 7 days. Then, on Day 7, when the ligature was removed, calcein was administered i.p. (20 mg/kg, Sigma Aldrich) along with anti-Sema4D mAb (3 mg/mouse, i.p.) or control IgG (3 mg/mouse, i.p.), and alizarin-3-methyliminodiacetic acid (30 mg/kg, i.p.) at Day 14, before sacrifice at Day 21. The schedule for the injection of calcein and Alizarin Red in relation to ligature removal and sacrifice is shown in Figure 3A. The maxillae were fixed in 4% formalin and dehydrated in ethanol followed by embedding in Technovit 2000 LC UV-curing acrylic resin. Sections were ground and polished to 60–80 μm by grinding equipment. The distance between the areas stained with calcein and Alizarin Red was measured using a laser scanning confocal microscope (Zeiss LSM 880). To evaluate a possible effect of anti-Sema4D mAb on systemic bone remodeling, mice (6- to 8-week-old C57BL/6j male, *n* = 5/group) received anti-Sema4D mAb (3 mg/mouse, i.p.) or control IgG (3 mg/mouse, i.p.), as well as calcein (20 mg/kg, i.p.) at Day 0, followed by alizarin-3-methyliminodiacetic acid (30 mg/kg, i.p.) at Day 7 and Calcein Blue (20 mg/kg, i.p.) at Day 13. Femurs were removed from the sacrificed mice at Day 14 for the above-noted resin-embedded fluorescence histology (Timeline: Figure 3C).

### 4.10. Statistical Analysis

Student’s *t* test was used for comparison of two different outcomes of experiments performed. Nonparametric data were evaluated using one-way or two-way ANOVA followed by Tukey’s post hoc analysis. *p* value < 0.05 was considered statistically significant.

## Figures and Tables

**Figure 1 ijms-23-05630-f001:**
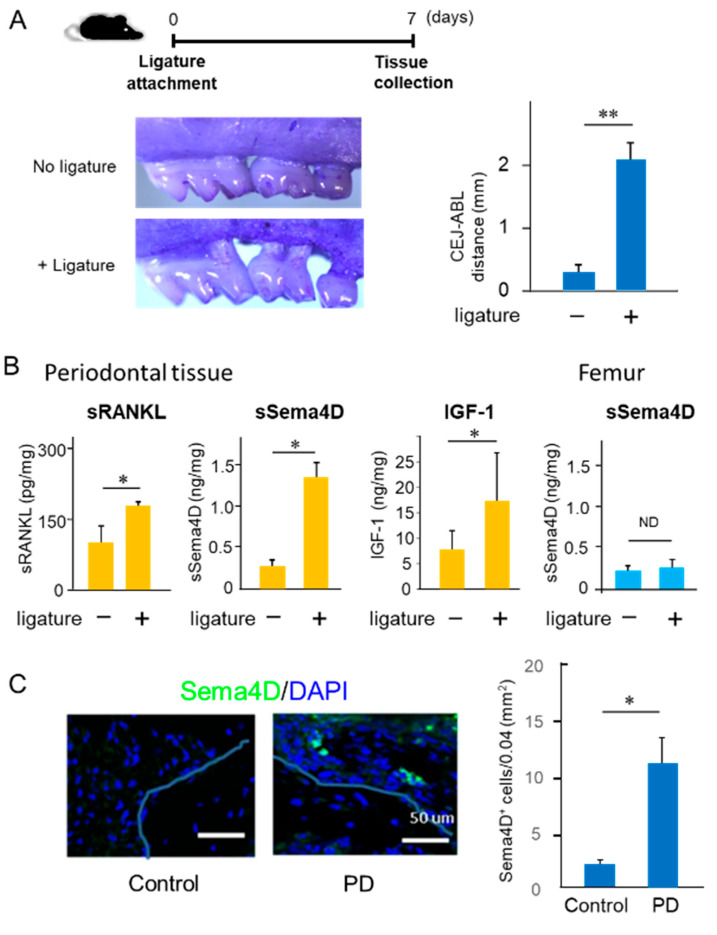
Semaphorin 4D (Sema4D) production in periodontal tissue of a ligature-induced periodontal disease (PD) mouse model. PD was induced by placing a ligature at maxilla second molars for 7 days. At the end point, each specimen was collected and used for experiments. (**A**) Alveolar bone loss, calculated as CEL-ABC distance of each group, is shown. (**B**) ELISA of sRANKL, sSema4D and IGF-1 in the homogenate of gingival tissue isolated around maxilla second molars, as well as ELISA of sSema4D in the homogenate of the femur and (**C**) immunohistochemical staining of Sema4D (green) and DAPI (blue) in periodontal tissue of ligature-induced periodontal disease (PD) and control mice. The line indicates alveolar bone surface. Scare bar = 50 μm. Values in the histogram are expressed as means ± S.D. *n* = 6/group. ** *p* < 0.01 or * *p* < 0.05 compared to control no-ligature group. ND: not detected.

**Figure 2 ijms-23-05630-f002:**
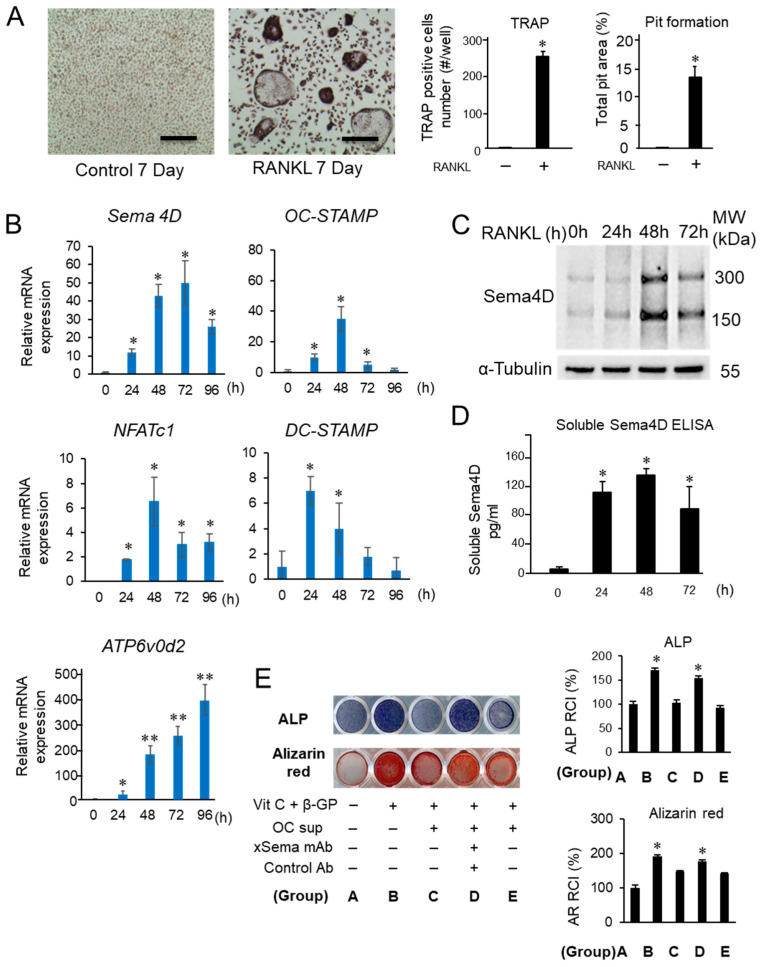
Expression of Sema4D during RANKL-induced osteoclastogenesis and the effects of Sema4D-containing supernatant from osteoclast culture on osteoblastogenesis. (**A**) Mouse bone marrow-derived mononuclear cells (BMMCs) cultured with M-CSF and RANKL for 7 days showed differentiation of TRAP+ multinuclear cells and elevated level of pit formation. Scale bar = 100 µm. (**B**) Expression of Sema4D mRNA, as well as mRNAs for osteoclastogenesis-related genes, including OC-STAMP, NFATc1 DC-STAMP and ATP6v0d2, was induced in BMMCs by stimulation with M-CSF and RANKL. (**C**) W-blot image of Sema4D expression by BMMCs stimulated with M-CSF/RANKL for 24, 48 and 72 h. Cell homogenates collected at respective incubation periods were lysed and subjected to SDS-PAGE under nonreducing conditions. (**D**) ELISA of Sema4D in supernatant from BMMCs cultured with RANKL/M-CSF for 24–72 h. Values are means ± S.D. *n* = 3/group. (**E**) MC3T3-E1 cells were cultured in basal medium in the presence or absence of supernatant from osteoclast precursor cells with or without OB genesis supplement or antibody shown in the figure. The photographs of ALP staining (top lane) and Alizarin Red staining (bottom lane) were captured after incubation at 7 and 21 days, respectively. The intensity of ALP staining and Alizarin Red staining in each group was calculated by Image J software and are presented in histograms (letters A–E shown in the histogram graphs correspond to letters assigned to groups in the photographs). Values are means ± S.D. *n* = 3/group. ** *p* < 0.01 or * *p* < 0.05 vs. BMMCs cultured in basal medium supplemented RANKL/M-CSF with/without anti-Sema4D mAb or control IgG for 7 days.

**Figure 3 ijms-23-05630-f003:**
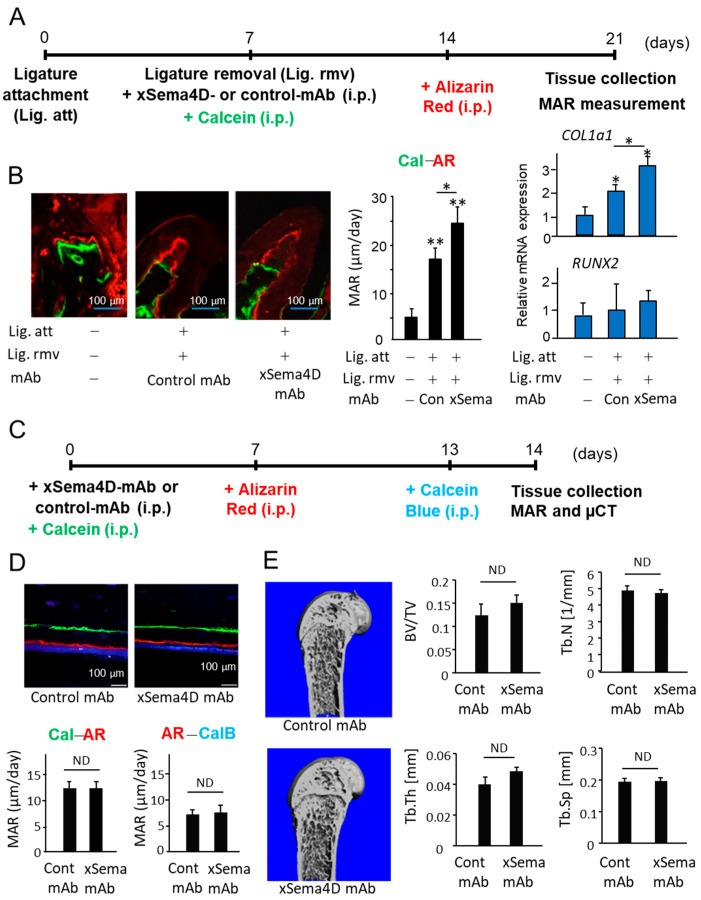
Effects of anti-Sema4D mAb on bone deposition occurring in PD-affected alveolar bone and intact healthy femur of mice. (**A**) Scheme of experimental timeline for ligature-induced PD, followed by removal of ligature and injection of calcein and Alizarin Red is presented. Anti-Sema4D mAb (3 mg/mouse, i.p.) or control IgG (3 mg/mouse, i.p.). (**B**) Representative photomicrographs of alveolar bone fluorescently labeled with Calcein (Cal: green) and Alizarin Red (AR: red) are shown. Scale bar = 100 μm. Mineral apposition rate (MAR) was calculated by the distance between the two fluorescent labels divided by the time between the two labeling periods. Expressions of *COL1a1* and *RUNX2* in the periodontal issue collected at Day 21 were measured using q-PCR. Values are means ± S.D. *n* = 6/group. ** *p* < 0.01 or * *p* < 0.05 vs. control no-ligature group. (**C**) Schematic of experimental timeline for femur bone-labeling of mice that received systemic anti-Sema4D-mAb or control mAb is shown. (**D**) Representative photomicrographs of fluorescently labeled femur and MAR are shown. Scale bar = 100 μm. Mineral apposition rate (MAR) was calculated by the distance between Calcein (Cal) and Alizarin Red (AR), as well as the distance between Alizarin Red (AR) and Calcein Blue (CalB). Values are means ± S.D. *n* = 5/group. (**E**) Micro-CT-based bone morphometry images taken 14 days after systemic administration of anti-Sema4D mAb or control mAb. Data obtained by micro-CT, including ratio of total bone volume (BV/TV), trabecular number (TbN), trabecular thickness (TbTh), and trabecular spacing (TbSp), are also shown. Values are means ± S.D. *n* = 5/group. ND: not detected.

**Figure 4 ijms-23-05630-f004:**
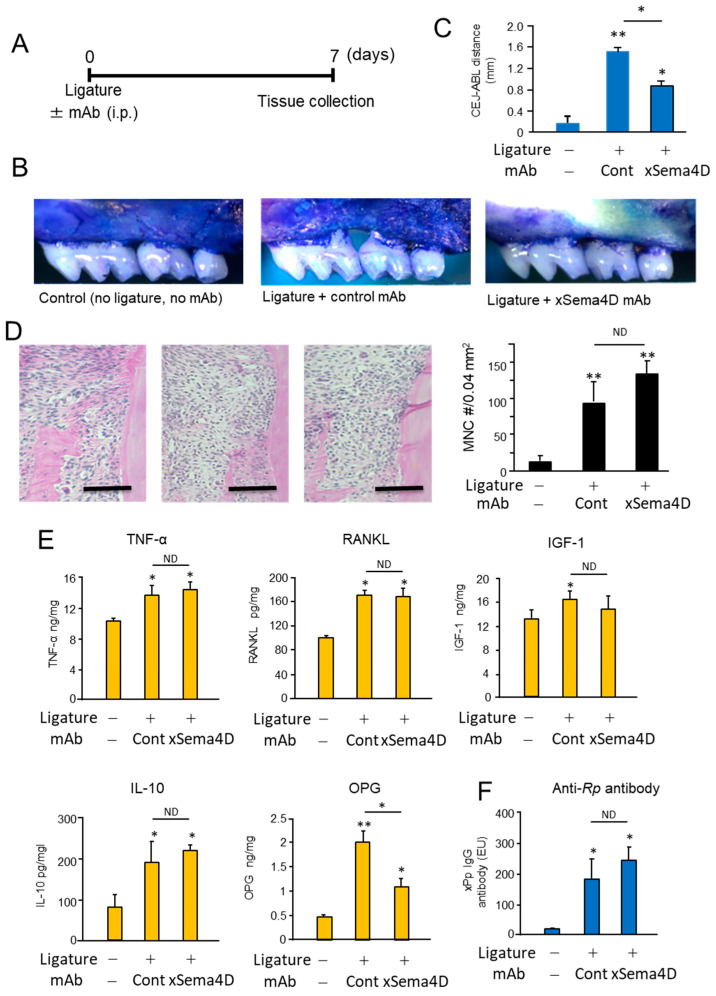
Possible role of locally produced Sema4D in the periodontal bone resorption and inflammatory responses in PD induced in mice by attachment of the ligature. To evaluate the effect of anti-Sema4D mAb on the alveolar bone loss induced in mice, the ligature was attached to the second molars of mice with or without systemic administration of anti-Sema4D mAb (3 mg/mouse) or control mAb (3 mg/mouse). After 7 days from the attachment of ligature, mice were euthanized for the collection of the respective tissue. (**A**) Scheme of experimental timeline for ligature-induced PD. Seven days after the attachment of ligature and injection with respective mAb, mice were euthanized, and jawbone was collected for the following post mortem analyses. (**B**) Periodontal bone loss: Images of alveolar bone resorption of maxilla each group. Representative images are shown. (**C**) CEL-ABC distance calculated for each group is expressed in a histogram. Values are means ± S.D. *n* = 6/group. * *p* < 0.05 vs. control no-ligature and no-injection group. (**D**) The histology section of periodontal tissue stained with H&E: inflammatory infiltrates in the ligature-induced periodontitis lesion as well as control tissue were measured. Scare bar = 100 μm. (**E**) ELISA of gingival tissue homogenate collected at Day 7 for detection of TNF-α, sRANKL, IGF-1, IL-10 and produced in the periodontal tissue. (**F**) Serum IgG antibody response to mouse oral opportunistic pathogen, *Rodentibacter pneumotropicus* (formally termed as *Pasteurella pneumotropica*), was measured using ELISA. Values are means ± S.D. *n* = 6/group ** *p* < 0.01 or * *p* < 0.05 vs. group without ligature and mAb injection. N.D., no difference between indicated groups.

**Figure 5 ijms-23-05630-f005:**
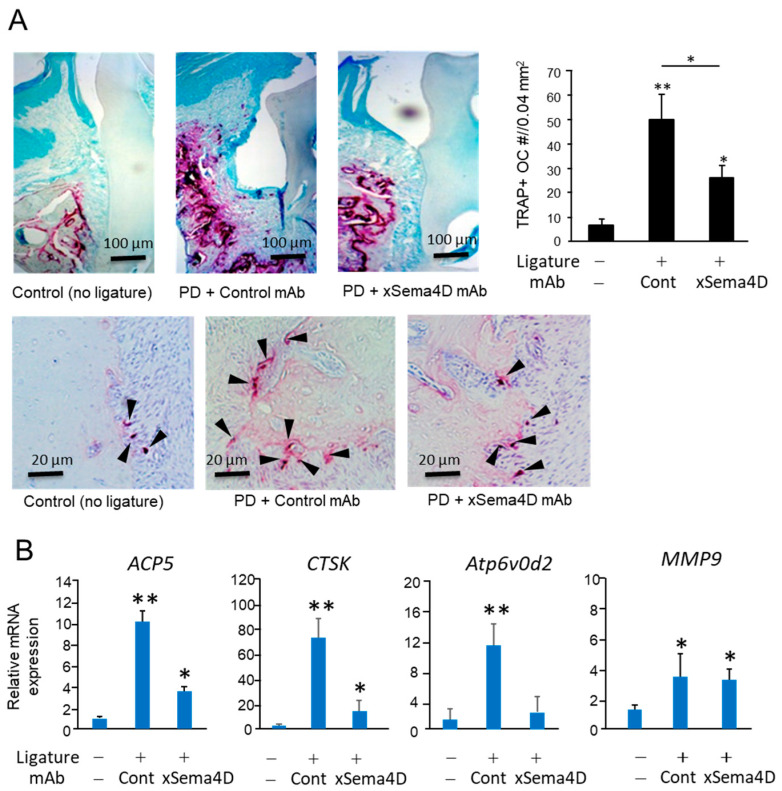
**Possible role of locally produced Sema4D in osteoclastogenesis in PD induced in mice by attachment of ligature.** After 7 days from the attachment of the ligature, three groups of mice received the same panel of treatment, as shown in Figure 4, and were euthanized for the collection of periodontal tissue. (**A**) Microscopic photograph of tartrate-resistant acid phosphatase (TRAP)-stained periodontal tissue section of each group. Black arrowhead indicates the TRAP-positive cell. The number of TRAP+ cells in each sample was calculated and expressed in a histogram. (**B**) The expressions of osteoclastogenesis-related genes, including *ACP5* (TRAP), *CTSK* (Cathepsin K), *Atp6v0d2* and *MMP9* in the periodontal tissue collected at Day 21 were determined using a real-time q-PCR. Values are means ± S.D. *n* = 6/group. * *p* < 0.05 and ** *p* < 0.01 vs. control no-ligature/no-mAb.

**Figure 6 ijms-23-05630-f006:**
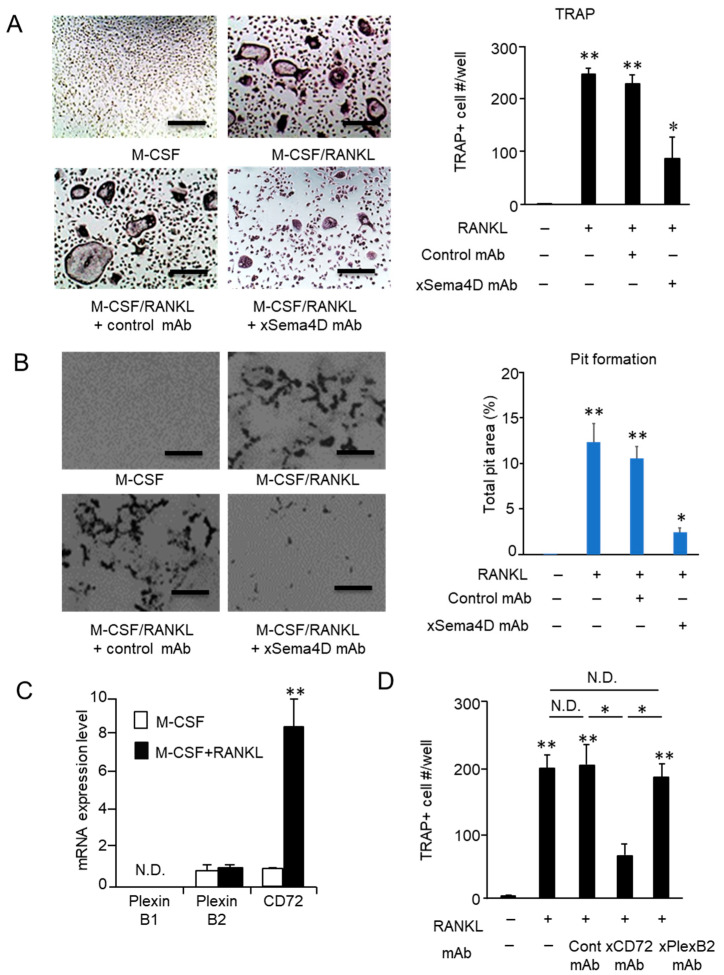
**Effect of sema4D produced by osteoclasts on the osteoclastogenesis and identification of Sema4D receptor expressed on osteoclast precursor cells.** To evaluate the effect of anti-Sema4D mAb on RANKL-induced osteoclastogenesis, BMMCs were stimulated with M-CSF/RANKL in the presence or absence of anti-Sema4D mAb (30 µg/mL) or control mAb (30 µg/mL) for 7 days. (**A**) Images of TRAP+ multinuclear cells differentiated in each group of BMMCs are shown. (**A**) The number of TRAP-positive multinuclear cells. Values are means ± S.D. *n* = 3/group. * *p* < 0.05, ** *p* < 0.01 vs. BMMCs cultured without RANKL/M-CSF or any mAb. Scale bar = 100 µm. (**B**) Images of TRAP+ multinuclear cells differentiated in each group of BMMCs are shown. (**B**) The area of resorbed pits was measured using ImageJ and expressed in a histogram. Values are means ± S.D. n = 3/group. * *p* < 0.05 vs. BMMCs cultured without RANKL/M-CSF or any mAb. Scale bar = 100 µm. (**C**) Expression level of mRNAs for PlexinB1, PlexinB2 and CD72 in BMMCs stimulated with RANKL/M-CSF for 48 h is shown. N.D., not detected. (**D**) BMMCs stimulated with M-CSF/RANKL in the presence or absence of anti-Plexin B2 mAb, anti-CD72 mAb or control mAb for 7 days. The number of TRAP-positive multinuclear cells (Day 7) in each group was calculated and is expressed in a histogram. Values are means ± S.D. *n* = 3/group * *p* < 0.05, ** *p* < 0.01 vs. BMMCs cultured without RANKL/M-CSF or any mAb, unless indicated by the dotted line. N.D., no difference between indicated groups.

## Data Availability

The data presented in this study are available on request from the corresponding author.

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
