# Peer review of "Locally Secreted Semaphorin 4D Is Engaged in Both Pathogenic Bone Resorption and Retarded Bone Regeneration in a Ligature-Induced Mouse Model of Periodontitis"

_ijms, 2022, doi:10.3390/ijms23105630_

Round 1

Reviewer 1 Report

Reviewer comments to the authors:

The current manuscript entitled “Locally secreted Semaphorin 4D is engaged in both pathogenic bone resorption and retarded bone regeneration in a ligature-induced mouse model of periodontitis” aimed to investigate the functional role of semaphorin 4D (Sema4D) in a mouse model of ligature-induced bone loss and the underlying molecular mechanisms in vitro. For this purpose, in vitro assay in BMM and MC3T3-E1 cells were carried out as well as in vivo studies in mice with experimental periodontitis (PD). The results indicated that Sema4D production was increased during PD progression and that systemic administration with anti-Sema4D mAb promoted bone regeneration and inhibited bone resorption. Moreover, anti-Sema4D mAb-mediated bone remodeling regulation was detected only in PD lesion, but not in homeostatic bone remodeling in the femur. The authors suggested that targeting Sema4D could be an interesting strategy for the development of regenerative approach for PD.

I found the study well structured, methodologically sounds, and the data are clear and compelling. Studies elucidating possible roles of cell-surface protein expressed in immune cells and its effects on periodontal tissues is one that needs to be more fully explored and I appreciate the time and efforts made to delve further into a subject matter with a possibility of adding to our body of knowledge. That being said, there are some comments and suggestions that the authors should address before resubmission, as described below:

- Please clarify why periodontal bone loss (BV/TV) was not measured by means of micro-CT? This method would provide a more accurate data analysis when compared to the bidimensional morphometric analysis (linear measurements). Bone mineral density evaluation would be interesting to measure.

- In figure 2, it is recommended to add an image regarding the pit formation on dentin slices to see the differences between groups. Moreover, I was wondering about the mRNA levels of osteoclast markers (c-Fos, ACP5, CTSK, MMP9). Did the authors checked for these markers?

- In figure 3. For the study of bone regeneration after ligature removal, is it know how the anti-Sema4D affect the levels of RUNX2, BMP, OC and OPN?

- In figure 4, I miss information about the inflammatory process in the periodontal tissues. It would be relevant to investigate inflammation by means of H&E stained slides. Perhaps a more objective method of analysis, i.e., immunostaining of inflammatory cells (CD31, CD3, CD45) would provide important insights in the inflammatory process, or to assess mRNA levels of IFN-y, TNF-a, IGF-1, IL-10, IL-4, IL-13, IL-17 in PD+anti-Sema4D mAb treated tissues compared to controls.

- It is recommended to increase the magnification of the images regarding the TRAP-stained section and adds some arrows to show the osteoclast cells (Fig. 4E). It is hard to see the stained cells even if significant enlargement of the image.

- Osteoprotegerin (OPG) is a known inhibitor of osteoclastogenesis. Is it known how anti-Sema4D mAb affects OPG levels?

- Please describe how the concentration of anti-Sema4D mAb (3 mg/mouse) was established as well as the dosage regime?

Yours sincerely,

Author Response

Please confirm attached.

Reviewer 2 Report

The manuscript “Locally secreted Semaphorin 4D is engaged in both pathogenic bone resorption and retarded bone regeneration in a ligature- induced mouse model of periodontitis” deal with investigation of Semaphorin 4D effect on bone in periodontitis induced mouse model.

In Author`s affiliation no. 7 is missing.

Abstract needs rough explanation of methods used with results (significances),

Overall, the manuscript is well written with adequate and also novel references used. Effect of Semaphorin 4D seems to be interestingly especially regarding medication related osteonecrosis of the jaw (MRONJ). As noted, more studies need to be performed in future. Did the authors conduct power analysis?

Author Response

Please confirm attached.

Reviewer 3 Report

This manuscript addresses the role of Sema4Din the process of periodontitis.  Most of the data shows clear differences, which would be interpreted without misinterpretation.

However I found some issues to be addressed.

1. How do you interpret the discrepancy between bone resorption of Sema4d–/– mice and antibody–treated periodontitis model mice: no difference in the former but low resorption in the latter.

2. xSema4D did not increase bone parameters although Sema4d–/– mice have petrotic bone (Negishi–Koga, Nat. Med., 2011).  This raises a suspect that increased bone of the periodontal tissue may not be caused by the direct effects on the bone cells.

3. Although periodontitis is an infectious disease, analyses of the immune system is missing (related to comment #2).

4. What is the source of IGF-1 in this experiment setting, degraded bone matrix, osteoblasts, or other cells?

5. xSema4D did not change TNF-a, RANKL or IGF-1 expression.  Because bone metabolism is not determined only by these 3 molecules, expression of other molecules should be addressed.

6. Effects of Sema4D on osteoclasts were described as the direct effects.  If so, downstream signal should be presented.  Also, it is possible that Sema4D activates MSCs included in the bone marrow cells to express RANKL.  This should be analyzed with much more  care.

7. Statistical analyses seem incorrect.  For nonparametric analyses, is  ANOVA really fit?  It is also problematic that one-way ANOVA is used in some figures where two-way ANOVA is suitable.

8. Reference should be revised such that the background is well explained to readers who are not specialists in this research field (e.g. refs. 7-9).

9. Proof of English should be done.

Author Response

Please confirm attached.

Round 2

Reviewer 1 Report

Dear authors,

Thank you for addressing all the points raised by the reviewer.

I feel that the manuscript improved significantly after the changes made according to the reviewers suggestions.

I can now recommend publication of the manuscript in its present form.

Yours sincerely,